# Patient-reported physical functioning and quality of life after pelvic ring injury: A systematic review of the literature

Hester Banierink[1]*, Kaj ten Duis[1], Klaus Wendt[1,2], Erik Heineman[3], Frank IJpma[1], Inge Reininga[1,2]

**1** Department of Trauma Surgery, University Medical Center Groningen, University of Groningen, Groningen, The Netherlands, **2** Emergency Care Network Northern Netherlands (AZNN), Northern Netherlands Trauma Registry, Groningen, The Netherlands, **3** Department of Surgery, University Medical Center Groningen, University of Groningen, Groningen, The Netherlands

* h.banierink@umcg.nl

## Abstract

### Background

Pelvic ring injuries are one of the most serious traumatic injuries with large consequences for the patients' daily life. During recent years, the importance of the patients' perception of their functioning and quality of life following injury has increasingly received attention. This systematic review reports on self-reported physical functioning and quality of life after all types of pelvic ring injuries.

### Methods

The online databases MEDLINE-PubMed and Ovid-EMBASE were searched for studies published between 2008 and 2019 to identify published evidence of patient-reported physical functioning and quality of life after which they were assessed for their methodological quality.

### Results

Of the 2577 articles, 46 were reviewed in full-text, including 3049 patients. Most studies were heterogeneous, with small cohorts of patients, a variety of injury types, treatment methods and use of different, often non-validated, outcome measures. The overall methodological quality was moderate to poor. Nine different PROMs were used, of which the Majeed Pelvic Score (MPS), SF-36 and EQ-5D were the most widely used. Mean scores respectively ranged from 75–95 (MPS), 53–69 (SF-36, physical functioning) and 0.63–0.80 (EQ-5D).

### Conclusions

Physical functioning and quality of life following pelvic ring injuries seem fair and tend to improve during follow-up. However, differences in patient numbers, injury definition,

**Data Availability Statement:** All relevant data are within the paper and its Supporting Information files.

**Funding:** The author(s) received no specific funding for this work.

**Competing interests:** The authors have declared that no competing interests exist.

treatment strategy, follow-up duration and type of PROMs used between studies hampers to elucidate the actual effects of pelvic ring injuries on a patient's life.

## Implications of key findings

Physicians and researchers should use valid and reliable patient-reported outcome instruments on large cohorts of patients with properly defined injuries to truly evaluate physical functioning and quality of life after pelvic ring injuries.

## Systematic review registration number

PROSPERO International prospective register of systematic reviews; registration number CRD42019129176.

## Introduction

Pelvic ring injuries can be seen as one of the most serious traumatic injuries with large consequences for the patients' daily life. Apart from the substantial mortality rates [1,2], principally in high-energy trauma, these injuries coincide with long periods of impaired mobilization and intense rehabilitation. In addition, pelvic ring injuries are increasingly caused by low-energy trauma in the frail elderly. Injury types vary from stable type A fractures, usually treated nonoperatively, to highly unstable type C fractures, often demanding operative fixation and long term recovery. Despite this, adequate prospective follow-up studies, both on short-term and long-term outcome, on pelvic ring injuries are lacking.

Many factors that characterise a patient's health status cannot be observed, measured with a device, or analysed with even the most sophisticated imaging methods. How a patient feels and performs remains largely impenetrable to devices [3]. The growing focus on patient-centred care has resulted in a shift in terms of outcome assessment and the increasing use of Patient-Reported Outcome Measurements (PROMs). These questionnaires seek to assess the influence of the patients' condition on their daily functioning and emotional status, and can provide critical information to enhance patient-centred health care [4]. Conceptually, PROMs can be viewed either as a 'tool for evaluation' or as a 'mechanism for improvement'.

No actual guidance exists for appropriate PROM-based assessment after pelvic ring injuries. Hence, the problem arises with regard to the long list of different PROMs used, many of which have no proof of being valid or reliable either. Lefaivre et al. [5] showed that many different types of generic outcome instruments as well as pelvis-specific measures are used to assess the outcomes after pelvic ring injuries. Besides, due to the wide variety in types of pelvic ring injuries and the variability in treatment strategy, outcomes are hard to compare, leaving physicians, researchers and patients in doubt about the actual outcomes following these injuries.

In this perspective, the main objective of the present systematic review was to identify and analyse published studies, thereby providing a representative overview of the outcomes in terms of patient-reported physical functioning and quality of life following pelvic ring injuries. Moreover, following the results of this review, our aim was to highlight whether changes can be made for future research in order to properly evaluate the consequences of these severe injuries.

## Methods

For this systematic review the PRISMA method [6] for literature collection and manuscript construction was followed. The review protocol has been registered in PROSPERO International prospective register of systematic reviews under registration number CRD42019129176.

### Identification of studies: Search strategy

The search strategy sought to retrieve references relating to physical functioning and quality of life after pelvic ring injuries. Therefore, the items "pelvis", "injury" and "outcome" were combined to develop the search strategy. Searches used medical subject headings (MeSH terms) and free text searching to combine terms specific to pelvic ring injuries with terms relevant to PROMs evaluation. The full electronic search strategy was developed in collaboration with an experienced medical librarian and is presented in Table 1. Two databases were searched to identify original articles: MEDLINE-PubMed (2008-15-04-2019) and Ovid-EMBASE (2008-15-04-2019).

### Inclusion and exclusion criteria and procedure

Eligible studies included patients aged 18 years or older with a pelvic ring injury. Studies that focused on the outcomes after nonoperative as well as operative treatment were eligible. The outcome measures used should include patient-reported outcome measures (PROMs). Except for case studies and conference abstracts, all study designs were accepted for inclusion. Concerning language, studies written in English, German, Spanish, French and Dutch were included. There was no limitation on the search by publication status. Studies on geriatric fractures or fragility fractures were excluded. Studies with a sample size of less than 20 patients in follow-up were excluded, because PROMs results based on so few patients seem unreliable. Moreover, studies that included outcomes after both pelvic ring injuries and acetabular fractures and that did not differentiate between these injuries in terms of outcomes, were excluded as well. The study selection was performed in two screening phases: 1) title and abstract

**Table 1. Search strings by database.**

| Database | Search string |
|---|---|
| MEDLINE-PubMed | ((("Pelvis"[Mesh:NoExp] OR "Sacrum"[Mesh] OR "Sacroiliac Joint"[Mesh] OR "Pubic Bone"[Mesh] OR "Pelvic Bones"[Mesh] OR pelvic[tiab] OR pelvis[tiab] OR sacrum[tiab] OR sacral[tiab] OR sacroiliac[tiab] OR pubic[tiab]) AND ("Wounds and Injuries"[Mesh] OR injur*[tiab] OR fractur*[tiab] OR trauma*[tiab]) AND ("Quality of Life"[Mesh] OR quality of life[tiab] OR "Recovery of Function"[Mesh] OR functional status[tiab] OR functional outcome*[tiab] OR physical function*[tiab] OR "Patient Outcome Assessment"[Mesh] OR patient reported outcome*[tiab] OR outcome assessment[tiab] OR SMFA[tiab] OR short musculoskeletal function assessment[tiab] OR EQ-5D[tiab] OR euroqol[tiab] OR SF-36[tiab] OR short form[tiab] OR SF-12[tiab] OR majeed[tiab] OR merle d'aubigne[tiab] OR (IPS[tiab] OR iowa[tiab])) NOT case reports[pt]) AND ("2008/01/01"[PDat]: "3000/12/31"[PDat])) |
| Ovid-EMBASE | ('pelvis'/de OR 'sacrum'/exp OR 'sacroiliac joint'/exp OR 'pubic bone'/exp OR 'pelvis fracture'/exp OR 'pelvis injury'/exp OR 'sacral fracture'/exp OR pelvic:ti,ab OR pelvis:ti,ab OR sacrum:ti,ab OR sacral:ti,ab OR sacroiliac:ti,ab OR pubic:ti,ab) AND ('injury'/exp OR injur*; ti,ab OR fractur*:ti,ab OR trauma*:ti,ab) AND ('quality of life'/exp OR 'convalescence'/exp OR 'patient-reported outcome'/exp OR 'patient outcome assessment':ti,ab OR 'patient reported outcome*':ti,ab OR 'quality of life':ti,ab OR 'functional status':ti,ab OR 'functional outcome*':ti,ab OR 'physical function*':ti,ab OR 'outcome assessment':ti,ab OR smfa:ti,ab OR 'short musculoskeletal function assessment':ti,ab OR 'eq 5d':ti,ab OR euroqol:ti,ab OR 'sf 36':ti,ab OR 'short form':ti,ab OR 'sf 12':ti,ab OR majeed:ti,ab OR (merle:ti,ab AND aubigne:ti,ab) OR (ips:ti,ab AND iowa:ti,ab)) AND [embase]/lim AND [2008–2018]/py NOT 'case report'/de NOT 'conference abstract'/it |

screening, and 2) full text screening. Both selection phases were independently performed by the same researchers (HB, IR).

## Data extraction

Data extraction was performed in sequence using a standardized data extraction spreadsheet developed prior to data extraction, for evaluating physical functioning and quality of life after pelvic ring injuries. During both selection phases, articles were selected on the basis of language, number of patients, age of patients, population (pelvic ring injury and human/non-human), study type and use of PROMs. Relevant data from the included articles were extracted by the senior author including the 1) names of the authors, 2) year of publication, 3) study design, 4) number of patients in follow-up, 5) type of pelvic injury, 6) details on type of treatment, 7) type of PROMs, and 8) outcome of PROMs. In case of discrepancies during any of the stages, the topic of disagreement was discussed within the entire review team (HB, IR, FIJ, KtD) in order to resolve disagreements.

## PROMs

The variables for which data were sought included all PROMs used to assess physical functioning and quality of life after pelvic ring injuries. These included the disease-specific Majeed Pelvis Score, Iowa Pelvic Score, Pelvic Outcome Score and Merle D'Aubigne-Postel score, as well as the generic Musculoskeletal Function Assessment, the Short Musculoskeletal Function Assessment, Short Form-36, Short Form-12 and EuroQuol-5D. A description of each of these PROMs can be found in S1 File.

## Assessment of methodological quality

Two authors (HB, IR) independently rated the methodological quality and risk of bias for each study by using a quality assessment tool developed by the McMaster University Occupational Therapy Evidence-Based Practice Research Group [7]. The Modified McMaster Critical Review form for Quantitive Studies consists of nine categories: citation, study purpose, literature, design, sample, outcomes, intervention, results, and conclusions and implications. This review form is appropriate to assess RCTs, cohort studies, single-case designs, before- and after-designs, case control studies, cross-sectional studies and case studies. The guidelines established by Law et al. [7] were utilized for the quality assessment. Every item was answered with 'yes; 1 point', 'no; 0 points', 'not addressed; 0 points' or 'not applicable (N/A); no points given'. The sum of these outcomes predicted the overall quality of the study assessed, ranging from 0 to 14 for RCTs and 0 to 12 for other study designs. The final score is given as the percentage of the maximum score. Qualitative assessment of intervention was not performed for the reason that this was irrelevant for the purpose of this review. Disagreements between the review authors were resolved through discussion until consensus was reached.

## Strategy for data synthesis

Data synthesis involved the comparison, combination, and summary of findings. Efforts were made to retrieve missing data on follow-up duration and missing scores on the questionnaires, by contacting the corresponding authors. Data is presented as part of a narrative synthesis, involving text and tables. The data are grouped according to the time of follow-up and the outcomes of the different types of PROMs that were used.

## Statistics

The results of the various questionnaires are shown according to the standards of the specific questionnaire, either as number with percentage or as mean with standard deviation or median with range or interquartile range (IQR). Pooled means and standard deviations were manually calculated for the complete cohort of every study in case the outcomes of the PROMs were provided for two or more groups.

## Results

### Selection of studies

The initial searches (conducted from January 2008 to April 15[th] 2019) generated 2577 articles. Following title and abstract assessment, 95 articles were reviewed in full text. A total of 46 articles were included in the review, of which most (N = 22) were cross-sectional studies, followed by case-control studies (N = 12), cohort studies (N = 10), one RCT and one combination of a cohort and cross-sectional study. Fig 1 demonstrates a flowchart of the inclusion procedure.

### Patient and injury characteristics

Overall, data of a total of 3049 patients were reported in the studies. The number of patients included in the studies varied widely, from as little as 20 patients [8] up to as much as 263 patients [9]. However, most studies were relatively small; only seven studies [9–15] included more than 100 patients and more than half reported on even less than 50 patients. Thirty-eight studies focused on unstable pelvic ring injuries (Type B and/or Type C according to the AO classification system [16]), whereas only six studies included all types of pelvic ring injuries [10,12,15,17–19]. Two studies focused on the outcomes after sacral fractures [20,21]. Both nonoperative treatment as well as several operative techniques were applied to treat the patients, although no study solely focused on the outcomes after nonoperative treatment. Operative techniques varied from external fixation to internal fixation with osteosynthesis plates to percutaneous fixation and other minimally invasive techniques. All included studies are described in Table 2.

### Methodological quality assessment

The results of the quality assessment of the included articles are presented in Table 3. Total scores in percentages ranged between 50% and 92%. The average score was 72%. No studies were excluded based on this assessment. Most studies scored fairly positive on the first four areas, regarding citation (1), study purpose (2), relevant background literature (3), and description of the sample (4). None of the studies justified sample size (5), which is the reason that no studies scored the maximum amount of points on the assessment. In the RCT [50] randomization of groups was performed (6), but it was not clearly described by which method (7). The first eight studies used valid (8) and reliable (9) PROMs, though some used both valid PROMs and PROMs of which the validity was not established (+/-). The ten studies in the list with the lowest quality scores did often not report results in terms of statistical significance (10) and did not use appropriate analysis methods (11). The last three areas regarding clinical importance (12), dropouts (13) and appropriate conclusions (14) were mostly sufficiently described.

### Patient-reported outcome measures

Thirty-eight studies [8,9,25,26,28–30,32–36,10,37–44,46,47,11,48–54,56,18–22,24] used a pelvic-specific PROM, either as a single instrument or in combination with a generic PROM.

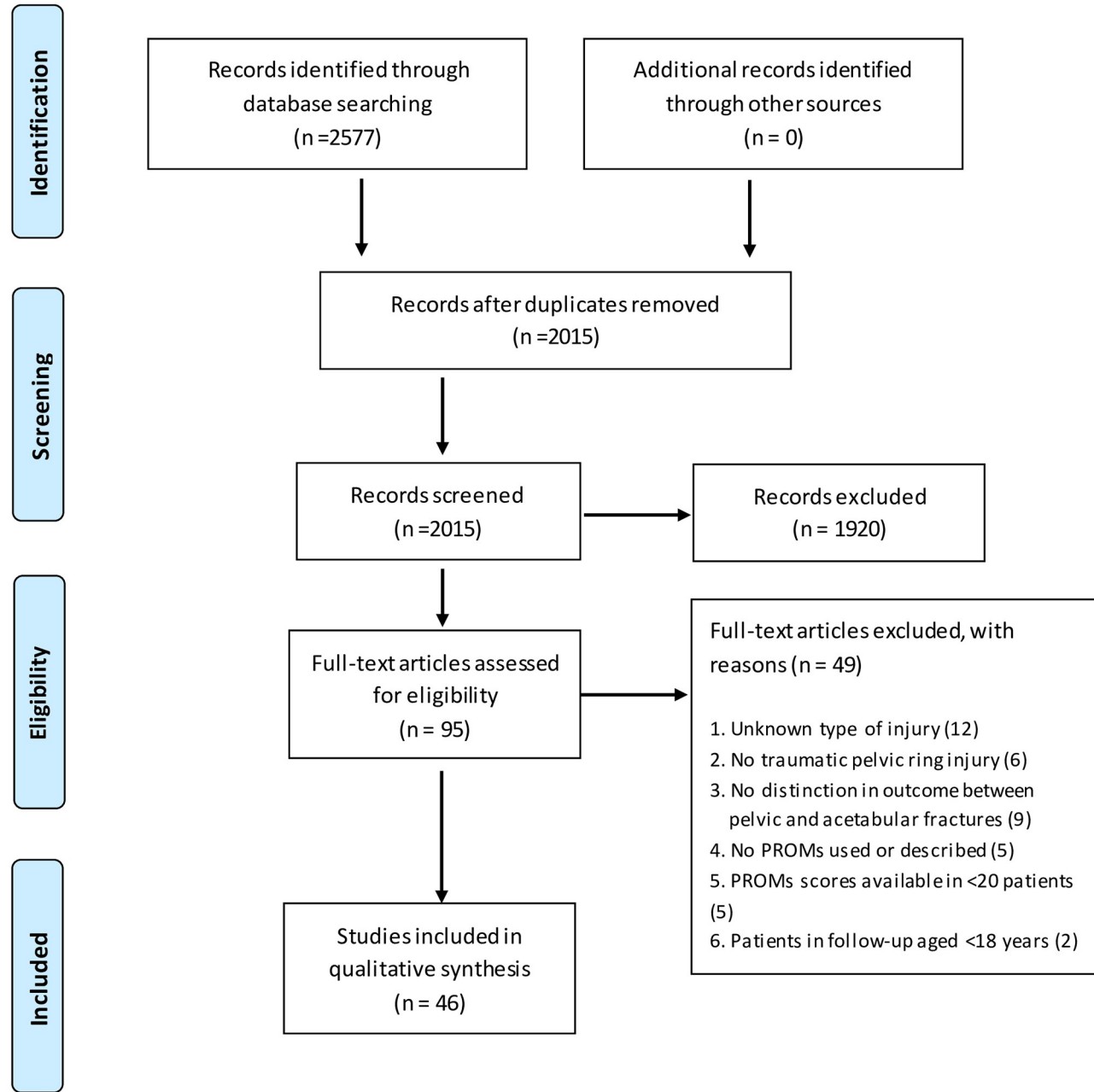

**Fig 1. Flow diagram according to the PRISMA method.**

Generic PROMs for physical functioning and quality of life were used in 15 studies [8,10,39,40,46,57,58,12,13,15,17,19,23,27,30]. The follow-up moment when these questionnaires were assessed ranged from six months to 15 years after the injury. Scores on the PROMs per study are given in Table 4.

## Patient-reported physical functioning

**PROMs results.** Of the 34 studies [8,9,26,28–30,32–37,10,39,41,43,44,46–51,11,52–54,19–22,24,25] that used the Majeed Pelvic Score (MPS), in 28 of them [9,11,32–38,41,43,44,20,47–

**Table 2. Study characteristics.**

| No. | Study | N | Method | Study period | Injury type (AO/OTA) [‡] | Interventions | PROMs | Follow-up in months |
|---|---|---|---|---|---|---|---|---|
| 1 | Abhishek et al. [22] | 41 | CS | 2007–2014 | B, C | Percutaneous ilio-sacral screw fixation | MPS | 12 |
| 2 | Adelved et al. [23] | 28 | CS/CSS | 1996–2001 | C | Surgical treatment with open or closed reduction | SF-36 | 12 (short FU) and 128 (mean; range 97–161) (long-term FU) |
| 3 | Ayvaz et al. [8] | 20 | CSS | 2004–2006 | B, C | Closed reduction and percutaneous fixation | SF-36, MPS, IPS, POS | 33 (mean; range 24–52) |
| 4 | Banierink et al. [12] | 192 | CSS | 2007–2016 | A, B and C | Nonoperative and operative treatment | SMFA-NL, EQ-5D | 53 (mean; range 12–120) |
| 5 | Bastian et al. [24] | 63 | CSS | 2004–2013 | B, C | Anterior fixation by modified Stoppa approach | MPS | 40 (mean; range 12–96) |
| 6 | Bi et al. [25] | 43 | CCS | 2012–2016 | B | S: Modified pedicle screw-rod fixation | MPS | 12 |
|   |   |   |   |   |   | C: Anterior pelvic external fixation |   |   |
| 7 | Borozda et al. [26] | 28 | CS | 2009–2013 | C | External fixation with separate anterior and posterior modules | MPS | 12 |
| 8 | Bott et al. [27] | 74 | CS | 1994–2005 | B, C | Surgical treatment | SF-36, EQ-5D | 180 (mean; range 132–264) |
| 9 | Brouwers et al. [10] | 195 | CSS | 2011–2015 | A, B and C | Nonoperative or surgical treatment | MPS, EQ-5D | 29 (mean; range 6–61) |
| 10 | Chen et al. [28] | 58 | CCS | 2002–2007 | C | S: Internal fixation with percutaneous reconstruction plate via posterior approach | MPS | 21 (mean; range 12–36) |
|   |   |   |   |   |   | C: Internal fixation with percutaneous sacroiliac screws via posterior approach |   |   |
| 11 | Chen et al. [29] | 21 | CSS | 2006–2009 | B | Endobutton technique for dynamic fixation of traumatic symphysis pubis disruption | MPS | 23 (mean; range 18–26) |
| 12 | Chen et al. [30] | 32 | CCS | 2002–2009 | B, C | S: Percutaneous iliosacral screw fixation | SF-36, MPS | 12 |
|   |   |   |   |   |   | C: Nonoperative treatment |   |   |
| 13 | Dienstknecht et al. [31] | 62 | CSS | 2000–2007 | C | Minimally invasive stabilizing system | POS | 37 (mean; range 36–42) |
| 14 | Feng et al. [32] | 26 | CCS | 2009–2013 | B | S: percutaneous fixation of traumatic pubic symphysis diastasis using a TightRope and external fixator | MPS | 15 (mean; range 12–20) |
|   |   |   |   |   |   | C: percutaneous cannulated screw fixation |   |   |
| 15 | Frietman et al. [19] | 37 | CSS | 2003–2013 | A, B and C | Symphyseal plating | SF-36, MPS | 34 (median; range 12–109) |
| 16 | Ghosh et al. [33] | 75 | CS | 2015–2016 | B, C | Nonoperative or surgical | MPS | 6 |
| 17 | Grubor et al. [18] | 47 | CSS | 1999–2009 | A, B and C | Nonoperative (sling, side-lying, resting) or Surgically (internal fixation, AO plates and screws) through Emile-Letournel's, suprapubic or sacroiliac approach | Merle d'Aubigne-Postel | ≥18 after trauma |
| 18 | Hoch et al. [14] | 128 | CS | 2004–2010 | B | Nonoperative and operative (minimally invasive posterior pelvic ring procedures). | SF-12, EQ-5D | 24 |
| 19 | Hoffman et al. [13] | 119 | CS | 2000–2010 | B | Nonoperative and operative (open or closed reduction and internal fixation) | SMFA | 6, 12 and 24 |
| 20 | Holstein et al. [15] | 172 | CSS | 2004–2011 | A, B and C | Nonoperative and operative | EQ-5D | 36 (median; range 12–72) |
| 21 | Hua et al. [34] | 23 | CSS | 2012–2015 | B, C | Minimally invasive interior internal pelvic fixator (INFIX) with or without a posterior pedicle screw-rod fixator | MPS | 14 (mean; range 6–27) |
| 22 | Kokubo et al. [35] | 82 | CSS | 1991–2010 | B, C | Nonoperative, external fixator or surgical | MPS | 12 (short FU) and 89 (mean; range 26–187) (long-term FU) |

*(Continued)*

**Table 2.** (Continued)

| No. | Study | N | Method | Study period | Injury type (AO/OTA) ‡ | Interventions | PROMs | Follow-up in months |
|---|---|---|---|---|---|---|---|---|
| 23 | Li et al. [36] | 64 | CCS | 2004–2006 | C | S: Surgical treatment with use of 3D printing model of the fracture | MPS | 12 and 144 |
| | | | | | | C: Without 3D printing model | | |
| 24 | Li et al. [37] | 47 | CSS | 2007–2014 | C | Iliac screw fixation in the posterior column of the ilium | MPS | 21 (mean; range 12–36) |
| 25 | Liu et al. [38] | 45 | CCS | 2016–2017 | B, C | S: Robot-assisted percutaneous screw placement combined with pelvic internal fixator | MPS | 5 (mean; range 4–12) |
| | | | | | | C: Percutaneous screw placement using conventional fluoroscopic imaging | | |
| 26 | Lybrand et al. [39] | 54 | CSS | 2000–2013 | B, C | Symphyseal fixation | EQ-5D, MPS | 84 (mean; range 24–168) |
| 27 | Ma et al. [9] | 263 | CCS | 2009–2015 | B, C | S: Internal fixation | MPS | 6 |
| | | | | | | C: External fixation | | |
| 28 | Muller et al. [40] | 36 | CS | 2004–2012 | C | Anterior subcutaneous internal fixator (ASIF) | SF-12, POS | 18 |
| 29 | Nie et al. [41] | 30 | CSS | 2015–2017 | B, C | Minimally invasive surgery assisted by 3D printing technology | MPS | 10 (mean; range 4–16) |
| 30 | Oh et al. [42] | 22 | CSS | 2008–2012 | B, C | Anterior plate fixation through Stoppa approach | Merle d'Aubigne-Postel | 16 (mean; range 10–51) |
| 31 | Park et al. [43] | 64 | CCS | 2009–2013 | B, C | S: ORIF with plate fixation and additional tension band wiring | MPS | 34 (mean; range 26–39) |
| | | | | | | C: ORIF with plate fixation alone | | |
| 32 | Schmitz et al. [17] | 55 | CSS | 2004–2014 | A, B and C | Nonoperative and operative fixation | SF-36, EQ-5D | 50 (mean; SD 35) |
| 33 | Schweitzer et al. [44] | 71 | CSS | 1998–2005 | B, C | Closed reduction and iliosacral percutaneous fixation | MPS | 31 (mean; range 12–96) |
| 34 | Shui et al. [11] | 117 | CSS | 2003–2013 | B, C | Percutaneous screw fixation | MPS | 14 (mean; range 6–24) |
| 35 | Vallier et al. [45] | 87 | CSS | 1997–2006 | B, C | Nonoperative, external or internal fixation | MFA | 41 (mean; range 16–137) |
| 36 | Van Loon et al. [46] | 32 | CSS | 1996–2008 | B | Nonoperative, external or internal fixation | SF-36, MPS | 84 (median) |
| 37 | Wang et al. [47] | 29 | CSS | 2010–2016 | B, C | Minimally invasive stabilization with pedicle screws connected to a transverse rod | MPS | 38 (mean; range 12–84) |
| 38 | Wang et al. [48] | 29 | CS | 2010–2016 | B, C | Modified pedicle screw-rod fixation | MPS | 12 |
| 39 | Wu et al. [49] | 23 | CS | 2013–2015 | B, C | Anterior fixation using a modified pedicle screw-rod fixator with or without posterior fixation using a transiliac internal fixator (TIFI) | MPS | 10 (mean; range 4–12) |
| 40 | Wu et al. [50] | 44 | RCT | 2009–2012 | B, C | S:Internal fixation through minimally invasive adjustable plate (MIAP) | MPS | S: 27 (mean; range 13–48) |
| | | | | | | C: internal fixation with locking compression plate (LCP) | | C: 22 (mean; range 12–42) |
| 41 | Yin et al. [51] | 74 | CCS | 2015–2017 | B, C | S: Anterior subcutaneous internal fixator (INFIX) | MPS | 27 (mean; range 21–32) |
| | | | | | | C: Plate fixation | | |
| 42 | Yu et al. [52] | 51 | CCS | - | B | S: reconstruction plate screw fixation | MPS | 29 (mean; range 18–54) |
| | | | | | | C: percutaneous cannulated screw fixation | | |
| 43 | Zhang et al. [20] | 42 | CCS | 2011–2017 | Unilateral sacral fractures | S: lumbopelvic fixation | MPS | 12 |
| | | | | | | C: Novel adjustable plate | | |
| 44 | Zhang et al. [53] | 22 | CSS | 2016–2017 | B | Nonoperative and operative | MPS | 12 (mean; range 8–15) |

(*Continued*)

**Table 2.** (Continued)

| No. | Study | N | Method | Study period | Injury type (AO/OTA) ‡ | Interventions | PROMs | Follow-up in months |
|-----|-------|---|--------|--------------|------------------------|---------------|-------|---------------------|
| 45 | Zhang et al. [21] | 70 | CCS | 2009–2016 | Unilateral zone II sacral fractures | S: Sacroiliac screw | MPS | 25 (mean; SD 5) |
| | | | | | | C: Minimally invasive adjustable plate | | |
| 46 | Zhu et al. [54] | 37 | CS | 2008–2012 | B, C | Ilioinguinal approach combined with a minimally invasive posterior approach | MPS | 12 |

* CSS, Cross-sectional study; RCT, randomized controlled trial; CS, cohort study; CCS, case-control study; S, study group; C, control group; IPS, Iowa Pelvic Score;

VAS, visual Analog scale; SF-36, MOS 36-item Short Form Health Survey; SF-12, Short Form-12; EQ-5D, EuroQuol-5D.

‡ The Young-Burgess classification was translated to the AO/OTA classification.

54,21,22,24–26,28,29] it was the only outcome instrument used. Most studies described the results in terms of the clinical grade. These were 'excellent' in 28–95% of the patients, 'good' in 5–64%, 'fair' in 0–25% and 'poor' in 0–19% of patients. Seven studies [19,38,39,46,50,51,53] only described the mean, ranging from 75 up to 95. Three studies [20,21,35] combined 'excellent' and 'good' results to 'satisfactory' (range 56–85%) and 'poor' and 'fair' to 'unsatisfactory' (range 15–37%). The Iowa Pelvic Score (IPS) was used by one study [8]. The mean score was 86 (range 82–90). The Pelvic Outcome Score (POS) was used by two authors [31,40]. The rates for 'excellent' in both studies were 29% and 31%, for 'good' 26% and 35%, for 'fair' 26% and 40% and for 'poor' 3% and 10%. Two studies [18,42] used the Merle D' Aubigne-Postel score for evaluation of function after pelvic ring injuries and graded it into 'excellent' (32% and 47%), 'good' (32% and 55%), 'fair' (9% and 13%), and 'poor' (0% and 12%). The Musculoskeletal Function Assessment (MFA) was used by only one study [45] evaluating female patients treated for pelvic ring injury. The mean score was 33 (SD 22). The Short Musculoskeletal Function Assessment (SMFA) was used in two studies [12,13]. One study [12] reported a score of 22 on the function index, 26 on the bother index and 21 on the lower extremity subscale. The other study [13] evaluated the scores of the SMFA on three time points (6, 12 and 24 months). Subsequently, scores on the function index were 28, 26 and 22, on the bother index 31, 30 and 24, and 33, 32 and 26 on the lower extremity subscale.

**Changes in physical functioning.** Three studies described physical functioning at different time points, and almost all of them showed improved scores at a later stage. Kokubo et al. [35] applied the MPS at one year and once again after a mean of 7.4 years, while Li et al. [36] also applied the MPS at one year and 10 years after the injury. Kokubo et al. found satisfactory (excellent + good) results of 63% after 1 year and 85% after 7.4 years. Unsatisfactory (fair + poor) results were found in 37% at one year and 15% after 7.4 years. Li et al. found excellent results in 60% after one year and 56% after 10 years, good results in 20 and 19%, fair in 20 and 25%, and no poor results. Hoffman et al. [13] used the SMFA at 6, 12 and 24 months revealing consecutive scores of 28, 26 and 22 on the function index, 31, 30 and 24 on the bother index and 33, 32 and 26 on the lower extremity subscale.

## Patient-reported quality of life

**PROMs results.** The SF-36 was used in seven studies [8,17,19,23,27,30,46]. Five studies [19,23,27,30,46] described all eight components of the SF-36 and one study [8] only described three of them. Scores ranged from 53 up to 69 (physical functioning), 24 to 71 (role physical), 49 to 68 (bodily pain), 42 to 65 (general health), 46 to 62 (vitality), 52 to 81 (social functioning), 49 to 85 (role emotional) and 52 to 78 (mental health). One study [17] only described the PCS and MCS score, which was 34 and 45 respectively. The SF-12 was used by two authors [14,40].

**Table 3. Scores of the quality assessment list ranged from best to worst score.**

| | No. | 1 | 2 | 3 | 4 | 5 | 6 | 7 | 8 | 9 | 10 | 11 | 12 | 13 | 14 | Total | % |
|---|---|---|---|---|---|---|---|---|---|---|---|---|---|---|---|---|---|
| Adelved et al. [23] | 1 | + | + | + | + | - | | | + | + | + | + | + | + | + | 11/12 | 92 |
| Banierink et al. [12] | 2 | + | + | + | + | - | | | + | + | + | + | + | + | + | 11/12 | 92 |
| Bott et al. [27] | 3 | + | + | + | + | - | | | + | + | + | + | + | + | + | 11/12 | 92 |
| Hoch et al. [14] | 4 | + | + | + | + | - | | | + | + | + | + | + | + | + | 11/12 | 92 |
| Hoffman et al. [13] | 5 | + | + | + | + | - | | | + | + | + | + | + | + | + | 11/12 | 92 |
| Holstein et al. [15] | 6 | + | + | + | + | - | | | + | + | + | + | + | + | + | 11/12 | 92 |
| Schmitz et al. [17] | 7 | + | + | + | + | - | | | + | + | + | + | + | + | + | 11/12 | 92 |
| Vallier et al. [45] | 8 | + | + | + | + | - | | | + | + | + | + | + | + | + | 11/12 | 92 |
| Brouwers et al. [10] | 9 | + | + | + | + | - | | | +/- | +/- | + | + | + | + | + | 10/12 | 83 |
| Chen et al. [30] | 10 | + | + | + | + | - | | | +/- | +/- | + | + | + | + | + | 10/12 | 83 |
| Frietman et al. [19] | 11 | + | + | + | + | - | | | +/- | +/- | + | + | + | + | + | 10/12 | 83 |
| Lybrand et al. [39] | 12 | + | + | + | + | - | | | +/- | +/- | + | + | + | + | + | 10/12 | 83 |
| Ma et al. [9] | 13 | + | + | + | + | - | | | - | - | + | + | + | + | + | 10/12 | 83 |
| Muller et al. [40] | 14 | + | + | + | + | - | | | +/- | +/- | + | + | + | + | + | 10/12 | 83 |
| Bastian et al. [24] | 15 | + | + | + | + | - | | | - | - | + | + | + | + | + | 9/12 | 75 |
| Feng et al. [32] | 16 | + | + | + | + | - | | | - | - | + | + | + | + | + | 9/12 | 75 |
| Kokubo et al. [35] | 17 | + | + | + | + | - | | | - | - | + | + | + | + | + | 9/12 | 75 |
| Liu et al. [38] | 18 | + | + | + | + | - | | | - | - | + | + | + | + | + | 9/12 | 75 |
| Park et al. [43] | 19 | + | + | + | + | - | | | - | - | + | + | + | + | + | 9/12 | 75 |
| Shui et al. [11] | 20 | + | + | + | + | - | | | - | - | + | + | + | + | + | 9/12 | 75 |
| Van Loon et al. [46] | 21 | + | + | + | + | - | | | +/- | +/- | - | + | + | + | + | 9/12 | 75 |
| Wang et al. [48] | 22 | + | + | + | + | - | | | - | - | - | - | + | + | + | 9/12 | 75 |
| Wang et al. [47] | 23 | + | + | + | + | - | | | - | - | + | + | + | + | + | 9/12 | 75 |
| Yin et al. [51] | 24 | + | + | + | + | - | | | - | - | + | + | + | + | + | 9/12 | 75 |
| Zhang et al. [20] | 25 | + | + | + | + | - | | | - | - | + | + | + | + | + | 9/12 | 75 |
| Wu et al. [50] | 26 | + | + | + | + | - | + | - | - | - | + | + | + | + | + | 10/14 | 71 |
| Bi et al. [25] | 27 | + | + | + | + | - | | | - | - | + | - | + | + | + | 8/12 | 67 |
| Borozda et al. [26] | 28 | + | + | + | + | - | | | - | - | + | + | + | - | + | 8/12 | 67 |
| Chen et al. [28] | 29 | + | + | + | + | - | | | - | - | + | + | + | - | + | 8/12 | 67 |
| Chen et al. [29] | 30 | + | + | + | + | - | | | - | - | + | - | + | + | + | 8/12 | 67 |
| Li et al. [36] | 31 | + | + | + | + | - | | | - | - | + | + | + | - | + | 8/12 | 67 |
| Li et al. [37] | 32 | + | + | + | - | - | | | - | - | + | + | + | + | + | 8/12 | 67 |
| Yu et al. [52] | 33 | + | + | + | - | - | | | - | - | + | + | + | + | + | 8/12 | 67 |
| Zhang et al. [20] | 34 | + | + | + | + | - | | | - | - | + | + | + | - | + | 8/12 | 67 |
| Dienstknecht et al. [31] | 35 | + | + | + | + | - | | | - | - | - | - | + | + | + | 7/12 | 58 |
| Grubor et al. [18] | 36 | + | + | - | + | - | | | - | - | + | + | + | - | + | 7/12 | 58 |
| Hua et al. [34] | 37 | + | + | + | + | - | | | - | - | - | - | + | + | + | 7/12 | 58 |
| Schweitzer et al. [44] | 38 | + | + | + | + | - | | | - | - | - | - | + | + | + | 7/12 | 58 |
| Wu et al. [55] | 39 | + | + | + | + | - | | | - | - | - | - | + | + | + | 7/12 | 58 |
| Zhang et al. [21] | 40 | + | + | + | + | - | | | - | - | + | - | + | - | + | 7/12 | 58 |
| Zhu et al. [54] | 41 | + | + | + | + | - | | | - | - | - | - | + | + | + | 7/12 | 58 |
| Abishek et al. [22] | 42 | + | - | + | + | - | | | - | - | - | - | + | + | + | 6/12 | 50 |
| Ayvaz et al. [8] | 43 | + | + | - | + | - | | | +/- | +/- | - | - | - | + | + | 6/12 | 50 |
| Ghosh et al. [33] | 44 | + | - | + | + | - | | | - | - | + | - | + | + | - | 6/12 | 50 |
| Nie et al. [41] | 45 | + | - | + | + | - | | | - | - | - | - | + | + | + | 6/12 | 50 |

(*Continued*)

**Table 3.** (Continued)

| | No. | 1 | 2 | 3 | 4 | 5 | 6 | 7 | 8 | 9 | 10 | 11 | 12 | 13 | 14 | Total | % |
|---|---|---|---|---|---|---|---|---|---|---|---|---|---|---|---|---|---|
| Oh et al. [42] | 46 | + | + | + | + | - | | | - | - | - | - | + | + | + | 6/12 | 50 |

Every plus sign means that the question was answered with 'yes'. Every minus sign means that a question was answered with 'no' or 'not addressed'. +/- was given in case both a valid as well as non-validated PROM was used and represents a score of 0.5. Questions 6 and 7 are only applicable for RCTs. The final two columns represent the total scores and percentages of maximal attainable scores (%).

The scores on the PCS were 37 and 43, and the scores on the MCS 43 and 46. The EQ-5D for the evaluation of quality of life was used in seven studies [10,12,14,15,17,27,39]. Mean scores ranged from 0.63 to 0.80.

**Changes in quality of life.** Only one study [23] assessed the SF-36 twice, at one year and once again after a mean of 10.7 years. Most of the scores improved after an interval of 10 years, although some decreased. Consecutive scores were as follows: physical functioning: 62 and 66, 42 and 46 (role physical), 51 and 49 (bodily pain), 65 and 59 (general health), 47 and 53 (vitality), 69 and 78 (social functioning), 62 and 49 (role emotional), 67 and 72 (mental health).

## Discussion

The management of and recovery of pelvic ring injuries has had gained attention over the years by clinicians and researchers. Although the focus primary laid on radiographic outcomes over the past decades, more recently this focus shifted towards the use of patient-reported outcome measures (PROMs). This is the first systematic review to evaluate outcomes in terms of physical functioning and quality of life after pelvic ring injuries. The extensive literature search resulted in the inclusion of 46 studies regarding patients with a broad range of injury types and treatment methods. Physical functioning and quality of life was mainly assessed between one and five years after pelvic ring injury. Most studies had small sample sizes, with more than half including even less than 50 patients. Besides, the quality of the studies was moderate to poor. Nine different outcome measures were used; 38 studies used disease-specific PROMs and 15 studies used generic PROMs. None of the disease-specific PROMs have been proven valid for use in patients with pelvic ring injuries. Overall, the recovery of physical functioning and quality of life following pelvic ring injuries seemed fair, although the reported results varied widely between studies and the different PROMs. Taking all of the above into account, it is challenging to conclude an overall result in terms of physical functioning and quality of life after pelvic ring injuries. Hence, some critical remarks can be made on the included studies based on the results of this systematic review.

Most studies reported on a wide variety of pelvic ring injury types. According to the AO/OTA classification system [16], pelvic ring injuries can be divided into type A, B or C injuries. However, sometimes the Young-Burgess classification [59] was used, which divides these injuries into 'anterior posterior compression (APC)', 'lateral compression (LC)' or 'vertical shear injuries (VS)'. In the studies that were included in this systematic review, it was not always clear what type of injury the patients had and most studies did not differentiate in the outcomes between for example B and C type injuries. Although type B as well as type C injuries are considered to be unstable fractures, type B injuries are simply rotationally unstable and therefore more likely to result in good outcomes, compared to the rotationally as well as vertically unstable type C injuries. Also, type A injuries were only assessed in six studies [10,12,15,17–19] even though this type consists most of all types of pelvic ring injuries [12]. Moreover, there was no differentiation in outcomes of patients with solely a pelvic ring injury,

**Table 4. Outcome of PROMs.**

| PROMs | Study | Year | N | Outcome of PROM at mean time of follow-up | | | |
|---|---|---|---|---|---|---|---|
| | | | | <12 months | 12–23 months | 24 months-5 years | >5 years |
| **MPS, N (%)** | | | | | | | |
| | Abhishek et al. [22] | 2015 | 41 | | Excellent: 21 (51), Good: 13 (32) Fair: 4 (10), Poor: 3 (7) | | |
| | Ayvaz et al. [8] | 2011 | 20 | | | Mean 93.3 (range 72–100) Excellent: 19 (95), Good: 1 (5) Fair: -, Poor: - | |
| | Bastian et al. [24] | 2016 | 63 | | | Excellent: 37 (59), Good: 12 (19) Fair: 9 (14), Poor: 5 (8) | |
| | Bi et al. [25] | 2017 | 43 | | Mean 81.97 (range 64–94) Excellent: 19 (44), Good: 17 (40) Fair: 7 (16), Poor: - | | |
| | Borozda et al. [26] | 2015 | 28 | | Mean 81 (range 58–97) Excellent: 12 (43), Good: 11 (39) Fair: 4 (14), Poor: 1 (4) | | |
| | Brouwers et al. [10] | 2018 | 195 | | | Mean 76 (SD 14.8) Excellent: 119 (61), Good 52 (27) Fair: 17 (9), Poor: 7 (3) | |
| | Chen et al. [28] | 2012 | 58 | | Mean: 80.7 Excellent: 19 (33), Good: 32 (55) Fair: 7 (12), Poor: - | | |
| | Chen et al. [29] | 2013 | 21 | | Excellent: 15 (71), Good: 5 (24) Fair: 1 (5), Poor: - | | |
| | Chen et al. [30] | 2012 | 32 | | Excellent: 10 (31), Good: 8 (25) Fair: 8 (25), Poor: 6 (19) | | |
| | Feng et al. [32] | 2016 | 26 | | Excellent: 18 (69), Good: 7 (27) Fair: 1 (4), Poor: - | | |
| | Frietman et al. [19]† | 2016 | 37 | | | Mean 75.3 (SD 19.5) | |
| | Ghosh et al. [33] | 2018 | 75 | Excellent: 27 (36), Good: 29 (39) Fair: 12 (16), Poor: 7 (9) | | | |
| | Hua et al. [34] | 2019 | 23 | | Excellent: 13 (57), Good: 6 (26) Fair: 4 (17), Poor: - | | |
| | Kokubo et al. [35] | 2017 | 82 | | Excellent + Good (satisfactory): 52 (63) Fair + Poor (Unsatisfactory): 30 (37) | | Excellent + Good (satisfactory): 70 (85) Fair + Poor (unsatisfactory): 12 (15) |
| | Li et al. [36] | 2017 | 64 | | Excellent: 38 (60), Good: 13 (20) Fair: 13 (20), Poor: - | | Excellent: 36 (56), Good: 12 (19) Fair: 16 (25), Poor: - |
| | Li et al. [37] | 2018 | 47 | | Mean 80.2 (range 48–100) Excellent: 13 (28), Good: 30 (64) Fair: 4 (8), Poor: - | | |
| | Liu et al. [38] | 2018 | 45 | Mean 85.4 (SD 8.9) | | | |
| | Lybrand et al. [39] | 2017 | 54 | | | | Mean 76 (SD 17) |
| | Ma et al. [9] | 2017 | 263 | Excellent: 125 (48), Good: 67 (25 Fair: 53 (20), Poor: 18 (7) | | | |
| | Nie et al. [41] | 2018 | 30 | Excellent: 21 (70), Good: 9 (30) Fair: -, Poor: - | | | |
| | Park et al. [43] | 2017 | 64 | | | Excellent: 31 (49), Good: 18 (28) Fair: 11 (17), Poor: 4 (6) | |

*(Continued)*

**Table 4.** (Continued)

| PROMs | Study | Year | N | <12 months | 12–23 months | 24 months-5 years | >5 years |
|---|---|---|---|---|---|---|---|
| | | | | Outcome of PROM at mean time of follow-up | | | |
| | Schweitzer et al. [44] | 2008 | 68 | | | Excellent + good: 62 (91) Fair: 4 (6), Poor: 2 (3) | |
| | Shui et al. [11] | 2015 | 117 | | Excellent: 48 (41), Good: 39 (33) Fair: 24 (21), Poor: 6 (5) | | |
| | Van Loon et al. [46] | 2011 | 32 | | | | Mean 95.7 |
| | Wang et al. [47] | 2017 | 29 | | | Excellent: 10 (35), Good: 16 (55) Fair: 3 (10), Poor: - | |
| | Wang et al. [48] | 2018 | 29 | | Excellent: 15 (52), Good: 12 (41) Fair: 2 (7), Poor: - | | |
| | Wu et al. [49] | 2018 | 23 | Excellent: 14 (61), Good: 7 (30) Fair: 2 (8), Poor: - | | | |
| | Wu et al. [50] | 2015 | 44 | | | Mean 81.7 (SD 8.4) | |
| | Yin et al. [51] | 2019 | 74 | | | Mean 86.2 (SD 7) | |
| | Yu et al. [52] | 2015 | 51 | | | Excellent: 36 (71), Good: 12 (24) Fair: 3 (5), Poor: - | |
| | Zhang et al. [20] | 2019 | 42 | | Excellent + Good (satisfactory): 33 (79) Fair + Poor (Unsatisfactory): 9 (21) | | |
| | Zhang et al. [53] | 2019 | 22 | | Mean 81 (SD 11) | | |
| | Zhang et al. [21] | 2019 | 70 | | | Excellent + Good (satisfactory): 56 (80) Fair + Poor (Unsatisfactory): 14 (20) | |
| | Zhu et al. [54] | 2015 | 37 | | Excellent: 29 (78), Good: 8 (22) Fair: -, Poor: - | | |
| **Iowa Pelvic Score (IPS)** | | | | | | | |
| | Ayvaz et al. [8] † | 2011 | 20 | | | Mean 86 (range 82–90) Excellent: 11 (55) Good: 9 (45) | |
| **Pelvic Outcome Score** | | | | | | | |
| | Dienstknecht et al. [31] # | 2011 | 62 | | | Excellent: 19 (31), Good: 16 (26) Fair: 25 (40), Poor: 2 (3) | |
| | Muller et al. [40] # | 2013 | 36 | | Excellent: 9 (29), Good: 11 (35) Fair: 8 (26), Poor: 3 (10) | | |
| **Merle d'Aubigne-Postel** | | | | | | | |
| | Grubor et al. [33] # | 2011 | 47 | | Excellent: 22 (47), Good: 15 (32) Fair: 4 (9), Poor: 6 (12) | | |
| | Oh et al. [42] # | 2015 | 22 | | Excellent: 7 (32), Good: 12 (55) Fair: 3 (13), Poor: - | | |
| **MFA** | | | | | | | |
| | Vallier et al. [45] † | 2012 | 87 | | | Mean: 33 (22) | |
| **SMFA** | | | | | | | |
| | Banierink et al. [12] | 2019 | 192 | | | Function index: 22 Bother index: 26 Lower extremity: 21 | |

(Continued)

**Table 4.** (Continued)

| PROMs | Study | Year | N | Outcome of PROM at mean time of follow-up | | | |
|---|---|---|---|---|---|---|---|
| | | | | <12 months | 12–23 months | 24 months-5 years | >5 years |
| | Hoffman et al. [13] | 2012 | 119 | Function index: 28 Bother index: 31 Lower extremity: 33 | Function index: 26 Bother index: 30 Lower extremity: 32 | Function index: 22 Bother index: 24 Lower extremity: 26 | |
| **SF-36** | | | | | | | |
| | Adelved et al. [23] † | 2014 | 28 | | PF 62 (28), RP 42 (45), BP 51 (32), GH 65 (23), VT 47 (20), SF 69 (27), RE 62 (43), MH 67 (25) | | PF 66 (26), RP 46 (45), BP 49 (29), GH 59 (26), VT 53 (23), SF 78 (22), RE 49 (44), MH 72 (21) |
| | Ayvaz et al. [8] | 2011 | 18 | | | BP: 3.3, GH: 4.4, SF: 7.9 | |
| | Bott et al. [27] † | 2019 | 74 | | PF 69 (30), RP 68 (32), BP 62 (28), GH 59 (28), VT 53 (23), SF 75 (29), RE 78 (31), MH 70 (23) | | |
| | Chen et al. [30] † | 2012 | 32 | | PF 53 (27), RP 24 (30), BP 50 (20), GH 42 (19), VT 46 (16), SF 52 (23), RE 50 (47), MH 52 (12) | | |
| | Frietman et al. [19] † | 2016 | 37 | | | PF 63 (26), RP 56 (41), BP 64 (27), GH 64 (25), VT 62 (30), SF 81 (24), RE 80 (32), MH 78 (18) | |
| | Schmitz et al. [17] † | 2018 | 55 | | | PCS: 34 (8) MCS: 45 (8) | |
| | Van Loon et al. [46] | 2011 | 32 | | | | GH: 62, VT: 58, MH: 72, BP: 68, SF: 80, RE: 85, RP: 71, PF: 74 |
| **SF-12** | | | | | | | |
| | Hoch et al. [14] * | 2016 | 128 | | | PCS 37 (11–56) MCS 43 (21–66) | |
| | Muller et al. [40] † | 2013 | 36 | | PCS 43 (2) MCS 46 (2) | | |
| **EQ-5D** | | | | | | | |
| | Banierink et al. [12] * | 2019 | 192 | | | Mean 0.76 (-.134–1) | |
| | Bott et al. [27] † | 2019 | 74 | | Mean 0.71 (SD 0.3) | | |
| | Brouwers et al. [10] † | 2019 | 195 | | | Mean 0.78 (0.26) | |
| | Hoch et al. [14] † | 2016 | 128 | | | Mean 0.75 (0.14) | |
| | Holstein et al. [15] ‡ | 2013 | 172 | | | Median: 0.78 (0.63–1.00) | |
| | Lybrand et al. [39] † | 2017 | 54 | | | | Mean 0.80 (0.20) |
| | Schmitz et al. [17] † | 2019 | 55 | | | Mean 0.63 (0.28) | |

* Data are given as N (%). Abbreviations:* Data given as mean (range).

† Data given as mean (SD).

‡ Data given as median (IQR).

# Data given as N (%). S, study group; C, control group; IPS, Iowa Pelvic Score; VAS, visual Analog scale; SF-36, MOS 36-item Short Form Health Survey; PF, physical functioning; RP, role physical; BP, bodily pain; GH, general health; VT, vitality; SF, social functioning; RE, role emotional; MH, mental health; SF-12, Short Form-12; EQ-5D, EuroQuol-5D; MFA, Musculoskeletal function assessment; SMFA, Short Musculoskeletal Function Assessment

and of patients with multiple injuries, which is seen in polytrauma patients. This may clearly affect results of generic PROMs.

None of the studies focused solely on the outcomes after nonoperative treatment of pelvic ring injuries. Only a few of the included studies [10,12,53,57,60,13–15,17,18,33,35,46] evaluated outcomes of patients that were treated either operatively or nonoperatively, while most studies only assessed operatively treated patients. Moreover, among the operatively treated patients, a wide variety of surgical techniques was used. The used techniques varied from external fixation, to purely anterior or posterior fixation, to a combination of both and even experimental techniques for specific pelvic ring injury types. Due to this variety in applied surgical techniques, which were often also poorly described, it was not possible to perform subgroup analyses. After all, the aim of this systematic review was to provide a general assessment of outcomes after pelvic ring injuries, but not of any specific operative approach.

Follow-up was mainly assessed between one and five years, missing the important short-term (<12 months) as well as long-term (>5 years) consequences of these injuries on the patients' daily life. Especially in the studies evaluating surgical techniques, the short-term follow-up is highly important, as this is a critical period in which the most improvement in physical functioning can be achieved. On the other hand, long-term follow-up might be just as important, revealing the late complications like gait impairment, chronic pelvic and back pain as well as delayed consequences of lumbosacral plexus injury [61]. Also, the unknown preinjury condition for physical functioning and quality of life leaves us guessing about the actual effect of the injury on the patients daily life.

Another problem in the evaluation of the studies was that the sample sizes of most studies were small, often including even less than 50 patients (N = 24). The methodological quality assessment revealed that no sample size calculation was performed in each of the studies, which makes it arguable whether enough patients were included to draw conclusions from in terms of physical functioning and quality of life. The quality assessment also revealed that, overall, the methodological quality was moderate and did not reach perfection in any of the studies, as all missed the justification for sample size. Moreover, many studies failed to achieve higher scores due to the use of nonvalidated outcome measures like the MPS.

The use of nine different PROMs was another issue. Of the four different disease-specific PROMs, the MPS was by far the most frequently used PROM in 34 studies, even though it has never been validated in patients with pelvic ring injuries. The reason for its frequent use could be explained by the compact length of the questionnaire and the possibility to compare outcomes to those of other studies. Similar to the results of this review, Lefaivre et al. showed that the MPS is the most commonly used pelvic outcome score [5]. Results were most often graded as 'excellent', although there was a wide variation in the proportion of patients that had an excellent score between the various studies. Only three studies [12,13,57] used two different generic PROMs (MFA and SMFA) to assess physical functioning, while quality of life was assessed in 13 different studies using the SF-12, SF-36 and EQ-5D, showing acceptable quality of life following pelvic ring injuries. The asset of these generic questionnaires is the availability of normative data to compare results with. A complicating factor was that the scores on identical questionnaires were often reported in different ways, making them hardly comparable. For example, the results on the MPS of the SF-36 were frequently reported by the categories (excellent, good etcetera), whereas other studies only presented mean scores with standard deviation, range, or a combination of these. In addition, scores varied widely, even between studies that used the same PROMs.

None of the disease-specific questionnaires that were used have been proven to be valid to assess physical functioning of patients with pelvic ring injuries, while all generic outcome instruments have. The ability of the outcomes of PROMs to improve decision-making in

clinical research relies on the psychometric strength of the instrument to capture the burden of disease or treatment. Reliability and validity are separate psychometric properties, both essential for any measure [62]. Measures can be highly reliable but not measure what they are supposed to measure [63]. Some studies compared pelvic-specific PROMs with generic PROMs to investigate the validity of disease-specific instruments in examining pelvic-specific areas, but failed to do so [5,64–66]. Hence, until there is a disease-specific questionnaire for pelvic ring injuries that is proven to be valid and reliable, it seems preferable to use a reliable and valid generic PROM to assess physical functioning and quality of life following these injuries. Another advantage of the latter is that, for these generic PROMs normative data often available is.

PROMs enable important clinical questions to be answered in clinical research [3]. Its use should be integrated in the clinical evaluation of a patient with pelvic ring injuries, next to the more objective measures like radiographic outcomes, because PROMs directly reflect the patients' perspective on the impact of their injury on daily life. Some types of pelvic injuries may look highly unfavourable on radiographic imaging, but the patient may grade his physical functioning and quality of life fairly well, or the other way around. Despite the fact that there has been discussion on the actual contribution of PROMs to the improvement of patient care, these instruments have the potential to facilitate patient involvement in treatment decision-making and provide guidance for health-care decisions [63]. Patients may monitor their health status over time and eventually will be more actively engaged in striving for health outcomes like full rehabilitation. Also, PROMs may help clinicians quickly identify which of their patients experience improved or deteriorated health outcomes. This may help to identify any structural patient complaints, which would suggest that refinements to care pathways might be needed. However, at this moment, PROMs function more as a tool for the use in clinical research, than they do in substantially changing medical practice.

## Strengths and limitations

Some strengths and limitations of this systematic review and its conclusions need to be addressed. To start with, this is the first systematic review to evaluate patient-reported physical functioning and quality of life after pelvic ring injuries. Also, search criteria were not limited by the type of study (e.g. cohort study, RCT), which provided a complete overview of all study results published during the past decade. Moreover, this systematic review underlines that some changes are needed in the future in order to examine the true consequences of pelvic ring injuries on the patients' daily life, for example to only use reliable and valid patient-reported outcome instruments. In this systematic review, a highly sensitive comprehensive search was conducted following the recommendations of an experienced medical librarian in order to identify articles of interest. For practical reasons though, only studies published in English, German, Spanish, French or Dutch were included in the final review, which might have led to selection bias. Additionally, studies published before 2008 were excluded after consultation with two experienced pelvic trauma surgeons. The argument for this was that, before 2008, treatment methods differed such an extent that including studies published before that time might lead to bias in the results of this systematic review. In this review, we included all types of pelvic ring injuries, treatment methods and types of PROMs. Due to this heterogeneity, individual outcomes of the included studies were not suitable for reliable comparisons. At last, sample sizes were not justified in any of the included studies.

## Conclusion

Even though the above-mentioned critical remarks make it ambitious to draw conclusions in terms of physical functioning and quality of life after pelvic ring injuries, the results imply that patients' physical functioning and quality of life seem reasonably fair and improve over time. However, a heterogeneous group of studies was presented, including small cohorts of patients with a wide range of injury types, treatment methods and diverse, often nonvalidated, outcome measures. Hence, there is a high need to use a valid and reliable outcome measure to evaluate and compare the recovery in terms of physical functioning and quality of life after pelvic ring injuries on large groups of patients. The following section provides some guidance for future research.

## Practical implications and recommendations for future research regarding use of PROMs after pelvic ring injuries

- Authors should clearly define the injury type according to the AO/OTA classification and distinguish between outcomes of different types of injuries. They should also distinguish between a pelvic ring injury as the only injury or as part of multiple injuries.

- Prospective longitudinal studies are needed with sufficient number of patients and multiple time intervals at short-term as well as long-term (>5 years) follow-up.

- (Recalled) pre-injury status of physical functioning and quality of life should be recorded.

- Only valid and reliable PROMs should be used, for example the SMFA for physical functioning and the EQ-5D or SF-36 for quality of life. These PROMs can be compared with age-specific norm data of the general population. The use of non-validated pelvic-specific PROMs should be avoided.

- There is still a challenging and a necessary task to validate existing pelvic-specific PROMs and develop an uniform PROM for pelvic injuries worldwide.

## Supporting information

**S1 File. Description of the included PROMs.**
(DOCX)

**S1 Checklist.**
(DOC)

## Acknowledgments

We thank Truus van Ittersum for helping with the search strategy.

## Author Contributions

**Conceptualization:** Hester Banierink.

**Data curation:** Hester Banierink.

**Formal analysis:** Hester Banierink, Inge Reininga.

**Investigation:** Hester Banierink.

**Methodology:** Hester Banierink.

**Supervision:** Kaj ten Duis, Klaus Wendt, Erik Heineman, Frank IJpma, Inge Reininga.

**Writing – original draft:** Hester Banierink.

**Writing – review & editing:** Frank IJpma, Inge Reininga.

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
