## [Decision Letter · Decision Letter 0]

20 Mar 2020

PONE-D-20-04766

Patient-reported physical functioning and quality of life after pelvic ring injury: a systematic review of the literature

PLOS ONE

Dear Ms. Banierink,

Thank you for submitting your manuscript to PLOS ONE. After careful consideration, we feel that it has merit but does not fully meet PLOS ONE’s publication criteria as it currently stands. Therefore, we invite you to submit a revised version of the manuscript that addresses the points raised during the review process.

The authors are required to respond to the reviewers comments.

We would appreciate receiving your revised manuscript by May 04 2020 11:59PM. To enhance the reproducibility of your results, we recommend that if applicable you deposit your laboratory protocols in protocols.io, where a protocol can be assigned its own identifier (DOI) such that it can be cited independently in the future. For instructions see: http://journals.plos.org/plosone/s/submission-guidelines#loc-laboratory-protocols

We look forward to receiving your revised manuscript.

Kind regards,

Osama Farouk

Academic Editor

PLOS ONE

Additional Editor Comments (if provided):

Thank you very much for submitting your article for consideration by PLOS ONE. The authors are required to respond to the reviewers comments.

Journal Requirements:

Reviewers' comments:

Reviewer's Responses to Questions

**Comments to the Author**

1. Is the manuscript technically sound, and do the data support the conclusions?

Reviewer #1: Yes

Reviewer #2: Yes

2. Has the statistical analysis been performed appropriately and rigorously? 

Reviewer #1: Yes

Reviewer #2: I Don't Know

3. Have the authors made all data underlying the findings in their manuscript fully available?

Reviewer #1: Yes

Reviewer #2: Yes

4. Is the manuscript presented in an intelligible fashion and written in standard English?

Reviewer #1: Yes

Reviewer #2: Yes

5. Review Comments to the Author

Reviewer #1: I would like to thank the authors for the well-conducted systematic review. I have only one comment, In terms of your results, what is the single most useful general and specific PROM you recommend to use in pelvic ring injuries

Reviewer #2: Although your manuscript involved a lot of papers in the literature, however it did not differentiate between patient groups. For example: are the papers included in the systematic review included polytrauma patients or not? Also are the papers included differentiate between geriatric pelvic fractures and other age groups?

As these 2 variables may affect the patient outcomes

6. PLOS authors have the option to publish the peer review history of their article (what does this mean?). If published, this will include your full peer review and any attached files.

Reviewer #1: No

Reviewer #2: No

---

## [Author Response · Author response to Decision Letter 0]

24 Mar 2020

Dear members of the editorial board, 

Thank you for your important remarks on our manuscript. We will address each of your points in the following section: 

1. In terms of your results, what is the single most useful general and specific PROM you recommend to use in pelvic ring injuries

Until there is a validated pelvic-specific PROM, we can only recommend to use a validated general PROM to evaluate the outcomes after these injuries (lines 391-394). We favour the SMFA for evaluation of physical functioning because it is compact, valid and reliable. For quality of life we recommend the EQ-5D or the SF-36 which are both valid and reliable, widely used, and often with available normative data to compare with.

2. Although your manuscript involved a lot of papers in the literature, however it did not differentiate between patient groups. For example: are the papers included in the systematic review included polytrauma patients or not? Also are the papers included differentiate between geriatric pelvic fractures and other age groups? As these 2 variables may affect the patient outcomes

Thank you for this remark. The included papers could also include patients with polytrauma, since we included all studies with patients over 18 years old with a pelvic ring injury. Most pelvic injuries are seen together with other injuries (with the exception of geriatric fractures), as the cause is often a high-energy trauma. These indeed affect the outcomes, but studies sadly did not differentiate between outcomes of polytrauma patients or patients with a pelvic ring injury as a single injury. This is a very important remark for future research and we added this in the manuscript as part of the discussion (lines 286-288) and of the recommendation (lines 386-387).

We excluded patients with geriatric fractures or fragility fractures of the pelvis, because the results in terms of PROMs do not seem comparable with the PROMs of patients with normal or good bone quality. This exclusion was not shown in the manuscript, therefore we added this in the in- and exclusion criteria (line 108).

We hope to have answered all of your questions and managed to correctly address all of your remarks in the revised manuscript. 

Hester Banierink, 

---

## [Decision Letter · Decision Letter 1]

17 Apr 2020

PONE-D-20-04766R1

Patient-reported physical functioning and quality of life after pelvic ring injury: a systematic review of the literature

PLOS ONE

Dear Ms. Banierink,

Thank you for submitting your manuscript to PLOS ONE. After careful consideration, we feel that it has merit but does not fully meet PLOS ONE’s publication criteria as it currently stands. Therefore, we invite you to submit a revised version of the manuscript that addresses the points raised during the review process.

We would appreciate receiving your revised manuscript by Jun 01 2020 11:59PM. To enhance the reproducibility of your results, we recommend that if applicable you deposit your laboratory protocols in protocols.io, where a protocol can be assigned its own identifier (DOI) such that it can be cited independently in the future. For instructions see: http://journals.plos.org/plosone/s/submission-guidelines#loc-laboratory-protocols

We look forward to receiving your revised manuscript.

Kind regards,

Osama Farouk

Academic Editor

PLOS ONE

Reviewers' comments:

Reviewer's Responses to Questions

**Comments to the Author**

1. If the authors have adequately addressed your comments raised in a previous round of review and you feel that this manuscript is now acceptable for publication, you may indicate that here to bypass the “Comments to the Author” section, enter your conflict of interest statement in the “Confidential to Editor” section, and submit your "Accept" recommendation.

Reviewer #1: All comments have been addressed

Reviewer #3: (No Response)

2. Is the manuscript technically sound, and do the data support the conclusions?

Reviewer #1: Yes

Reviewer #3: Yes

3. Has the statistical analysis been performed appropriately and rigorously? 

Reviewer #1: Yes

Reviewer #3: Yes

4. Have the authors made all data underlying the findings in their manuscript fully available?

Reviewer #1: Yes

Reviewer #3: Yes

5. Is the manuscript presented in an intelligible fashion and written in standard English?

Reviewer #1: Yes

Reviewer #3: Yes

6. Review Comments to the Author

Reviewer #1: (No Response)

Reviewer #3: General comments:

In general, pelvic ring injuries are one of the most serious traumatic injuries, the research idea is significantly important and is interesting to be presented in a systematic review study about patient-reported physical functioning and quality of life after pelvic ring injury with large consequences for the patients’ daily life.

The manuscript is generally well written and structured. The study has many limitations, but in general it highlighted on valuable information. There is a concern to include all pelvic ring fractures with all degrees and type of treatment. In addition, this study included all tools of PROMs measurements However, I have provided some remarks below.

Methods:

-Page 11, line 95: please write inclusion instead of “In.”

Results:

- All case either treated with operative or non operative treatment ways were included, there are no data presented to compare between those types of patients regarding PROMs

- page13, line 157: in the sentence (conducted January 2008 to April 15th 2019) write “from” after conducted.

- line169: write the reference after “ AO classification system”

It’s better to insert the tables of results in the main manuscript better than additional files, as they are an integral part of the manuscript

Limitations:

- please include the non-calculation of the sample size in all studies.

- Also, gathering of data with multiple tools of assessment is a major limitation.

7. PLOS authors have the option to publish the peer review history of their article (what does this mean?). If published, this will include your full peer review and any attached files.

Reviewer #1: No

Reviewer #3: Yes: Dalia G Mahran

---

## [Author Response · Author response to Decision Letter 1]

20 Apr 2020

Dear members of the editorial board, 

Thank you for your important remarks on our manuscript. We will address each of your points in the following section: 

Methods section

1. Page 11, line 95: please write inclusion instead of “In.”

Done

Results section

1. All case either treated with operative or non-operative treatment ways were included, there are no data presented to compare between those types of patients regarding PROMs

This is right. In this study we wanted to provide an overview of all available outcomes and not compare types of treatment. Of course this would be interesting, but with all the different operative techniques that were used, as well as outcome measures, the results of this comparison would not be valid. 

2. Page 13, line 157: in the sentence (conducted January 2008 to April 15th 2019) write “from” after conducted.

Done

3. line169: write the reference after “ AO classification system”

Done

4. It’s better to insert the tables of results in the main manuscript better than additional files, as they are an integral part of the manuscript

We agreed and added the tables in the main manuscript

Limitations section

1. Please include the non-calculation of the sample size in all studies.

Done

2. Also, gathering of data with multiple tools of assessment is a major limitation.

This is also added in the limitations section, together with the fact that it includes all types of pelvic ring injuries and treatment methods. 

We hope to have answered all of your questions and managed to correctly address all of your remarks in the revised manuscript. 

Hester Banierink, 

---

## [Decision Letter · Decision Letter 2]

1 May 2020

Patient-reported physical functioning and quality of life after pelvic ring injury: a systematic review of the literature

PONE-D-20-04766R2

Dear Dr. Banierink,

We are pleased to inform you that your manuscript has been judged scientifically suitable for publication and will be formally accepted for publication once it complies with all outstanding technical requirements.

With kind regards,

Osama Farouk

Academic Editor

PLOS ONE

Additional Editor Comments (optional):

Reviewers' comments:

Reviewer's Responses to Questions

**Comments to the Author**

1. If the authors have adequately addressed your comments raised in a previous round of review and you feel that this manuscript is now acceptable for publication, you may indicate that here to bypass the “Comments to the Author” section, enter your conflict of interest statement in the “Confidential to Editor” section, and submit your "Accept" recommendation.

Reviewer #3: (No Response)

2. Is the manuscript technically sound, and do the data support the conclusions?

Reviewer #3: Yes

3. Has the statistical analysis been performed appropriately and rigorously? 

Reviewer #3: Yes

4. Have the authors made all data underlying the findings in their manuscript fully available?

Reviewer #3: (No Response)

5. Is the manuscript presented in an intelligible fashion and written in standard English?

Reviewer #3: Yes

6. Review Comments to the Author

Reviewer #3: (No Response)

7. PLOS authors have the option to publish the peer review history of their article (what does this mean?). If published, this will include your full peer review and any attached files.

Reviewer #3: No

---

## [Editor Report · Acceptance letter]

6 Jul 2020

PONE-D-20-04766R2 

Patient-reported physical functioning and quality of life after pelvic ring injury: a systematic review of the literature 

Dear Dr. Banierink:

I'm pleased to inform you that your manuscript has been deemed suitable for publication in PLOS ONE. Congratulations! Your manuscript is now with our production department. 

Kind regards, 

on behalf of

Dr. Osama Farouk 

Academic Editor

PLOS ONE